# Cellular, Molecular and Clinical Aspects of Aortic Aneurysm—Vascular Physiology and Pathophysiology

**DOI:** 10.3390/cells13030274

**Published:** 2024-02-01

**Authors:** Dominika Domagała, Krzysztof Data, Hubert Szyller, Maryam Farzaneh, Paul Mozdziak, Sławomir Woźniak, Maciej Zabel, Piotr Dzięgiel, Bartosz Kempisty

**Affiliations:** 1Division of Anatomy, Department of Human Morphology and Embryology, Wroclaw Medical University, 50-368 Wroclaw, Poland; dominika.domagala@umw.edu.pl (D.D.); krzysztof.data@umw.edu.pl (K.D.); hubert.szyller@student.umw.edu.pl (H.S.); slawomir.wozniak@umw.edu.pl (S.W.); 2Fertility, Infertility and Perinatology Research Center, Ahvaz Jundishapur University of Medical Sciences, Ahvaz, Iran; maryamfarzaneh2013@yahoo.com; 3Prestage Department of Poultry Science, North Carolina State University, Raleigh, NC 27607, USA; pemozdzi@ncsu.edu; 4Physiology Graduate Faculty, North Carolina State University, Raleigh, NC 27613, USA; 5Division of Histology and Embryology, Department of Human Morphology and Embryology, Wroclaw Medical University, 50-368 Wroclaw, Poland; maciej.zabel@umw.edu.pl (M.Z.); piotr.dziegiel@umw.edu.pl (P.D.); 6Division of Anatomy and Histology, University of Zielona Góra, 65-046 Zielona Góra, Poland; 7Department of Physiotherapy, University School of Physical Education, 51-612 Wroclaw, Poland; 8Institute of Veterinary Medicine, Nicolaus Copernicus University, 87-100 Torun, Poland; 9Department of Obstetrics and Gynecology, University Hospital and Masaryk University, 602 00 Brno, Czech Republic

**Keywords:** inflammation, vessel, VSMCs, ECM

## Abstract

A disturbance of the structure of the aortic wall results in the formation of aortic aneurysm, which is characterized by a significant bulge on the vessel surface that may have consequences, such as distention and finally rupture. Abdominal aortic aneurysm (AAA) is a major pathological condition because it affects approximately 8% of elderly men and 1.5% of elderly women. The pathogenesis of AAA involves multiple interlocking mechanisms, including inflammation, immune cell activation, protein degradation and cellular malalignments. The expression of inflammatory factors, such as cytokines and chemokines, induce the infiltration of inflammatory cells into the wall of the aorta, including macrophages, natural killer cells (NK cells) and T and B lymphocytes. Protein degradation occurs with a high expression not only of matrix metalloproteinases (MMPs) but also of neutrophil gelatinase-associated lipocalin (NGAL), interferon gamma (IFN-γ) and chymases. The loss of extracellular matrix (ECM) due to cell apoptosis and phenotype switching reduces tissue density and may contribute to AAA. It is important to consider the key mechanisms of initiating and promoting AAA to achieve better preventative and therapeutic outcomes.

## 1. Introduction

Abdominal aortic aneurysm (AAA) is a clinically challenging and potentially lethal disease that often escapes early detection and remains a substantial danger to the aging population [1]. The disease is characterized by the local enlargement and swelling of the abdominal aorta, leading to a rupture of the aortic wall. It constitutes a significant challenge for modern medicine and requires special attention from both clinical doctors and researchers [2]. The initiation and development of AAA often is completely asymptomatic, while the risk of severe consequences is constantly increasing [3]. Most cases are diagnosed by the patient exhibiting aortic dissection or rupture [4]. The epidemiology of AAA combined with its association with atherosclerosis and other cardiovascular diseases (CVDs) underscores its importance as a public health problem [5]. The prevalence of AAA in the general population is estimated to be dependent on gender and age, with its occurrence being most prevalent in elderly men. AAA has a complex multifactorial etiology, including inflammation, genetic factors or single nucleotide variants (SNV) of individual genes, as matrix metalloproteinase-2 (MMP-2), matrix metalloproteinase-3 (MMP-3), matrix metalloproteinase-13 (MMP-13) [6], interleukin-6 (IL-6) and tumor necrosis factor α (TNF-α) [7]. These genes influence intra- and extra-environmental interactions, including signaling pathways [8]. Another crucial factor is the weakening and stiffening of the aortic wall, which may be influenced by the structure and content of collagen, elastin and the other extracellular matrix (ECM) components [9,10]. Metabolism and the viability of vascular cells are also crucial for aorta resistance and physiology. The loss of vascular smooth muscle cells (VSMCs) and fibroblasts, due to the function of ECM protein synthesis, with a high activity of proteases, lead to aortic wall degradation [11,12]. Therefore, understanding the complex mechanisms governing the development and progression of AAA, along with advances in diagnostic methods and therapeutic interventions, is of great importance to public health and clinical practice.

## 2. Epidemiology

Aortic aneurysm is a non-physiological extension of the aorta, which may occur in the thoracic or abdominal section. Thus, it is important to distinguish between AAA, which occurs more often, and thoracic aortic aneurysm (TAA), which develops less frequently [13]. AAA is defined as an extension of the aorta exceeding 3 cm in transverse diameter. The estimated prevalence of AAA in men over 60 years of age is approximately 4–8%, while in women it is 0.5–1.5%. AAA is less common in women in the general population; however, this may be due to the fact that AAA in women is still underdiagnosed and undertreated. However women with AAA are now four times more likely to have a ruptured AAA compared to men of the same age and account for one third of deaths due to AAA ruptures [14]. Van de Luijtgaarden et al. [15] examined patients with AAA invited to the multidisciplinary vascular/genetics outpatient clinic between 2009 and 2012 for an assessment of family history using questionnaires. The risk of this disease for male and female relatives was calculated separately and stratified by the sex of the AAA patients. They showed that AAA was significantly more common among the relatives of female AAA patients compared to the relatives of male cases. Moreover, it is worth mentioning that AAA in women has a faster growth rate and ruptures at smaller diameters [15]. The main non-genetic factor that contributes to AAA is cigarette smoking, whereas the genetic factors include male gender, involutional changes (advanced age) and genetic predisposition [16]. The most dangerous complication of AAA is its rupture. The frequency of ruptured AAAs is 5.6–17.5 per 100,000 in Western countries, and the worldwide mortality of patients with a ruptured AAA is estimated at 80–90% [17]. A recent epidemiological study conducted in Poland suggests that the occurrence of AAA is as common as in other European countries. However, the prevalence of AAA may be correlated with an insufficient diagnosis of aneurysm resulting from the absence of regular preventative screening programs [18].

## 3. Anatomical Structure

The aortic wall is built of three layers: outer or adventitia, middle or media and inner or intima (Figure 1). The intima is composed of single endothelial cells, which are located on areolar connective tissue. The intima plays an important role in intercellular signaling, and it contributes to the expression of proinflammatory cytokines, which can modulate immune cell activity [19]. The media is composed of smooth muscle, elastin and collagen fibers which provide aorta elasticity and contractility. The adventitia is constituted mainly of fibroblasts, which produce ECM fibers. The adventitia provides strength and prevents the excess stretching of the aortic wall. In this layer there are also vascular vessels (Latin: vasa vasorum) which are predisposed to adventitial inflammation [20]. The superficial layer of all types of vessels, except cerebral and pulmonary, is covered with a layer of perivascular adipose tissue (PVAT). In physiology conditions, the role of PVAT is to take part in regulating vascular tension, thermoregulation and the releasing of molecules influencing metabolism and the proliferation of VSMCs [21]. An overgrown PVAT layer is suggested as a direct factor responsible for the development of obesity-related CVDs [22].

## 4. Tissue and Cellular Structure and Main Abnormalities of Aorta

The ascending and descending sections of the aorta differ in origin. The VSMCs of the ascending aorta come from the neural crest, while the cells of the descending aorta are somitic mesoderm-derived [23]. The border between them consists of a potential location of some CVDs, such as the coarctation of the aortic arch, an interrupted aortic arch and aortic arch aneurysm [24]. The differences in the physiology of those sections may also be a reason for the disproportion between TAA and AAA incidence [25]. Due to high blood pressure, dissecting aneurysms are much more common in the thoracic aorta than the abdominal aorta [26]. In the morphology of blood vessels, especially the greatest artery of the body, the appropriate cooperation and functional complementation of the three layers of the aorta are crucial. Disturbances in the communication of aortic layers, like depositions of atherosclerotic plaque between layers, may be one of the factors contributing to the development of AAA. Atherosclerotic plaque consequently reduces the lumen of the vessels and significantly impedes blood flow through it [27]. As a result, AAA may be formed as a reaction to histopathological changes. The narrowing of the aortic lumen causes compensatory changes in the aortic wall, and consequently, ECM remodeling occurs to normalize the vessel lumen diameter. Rapidly occurring remodeling significantly reduces the organized structure of individual layers, weakening the strength and elasticity of the vessels [28]. However, further research is needed regarding the above relationship. Aortic layers differ in function, which is the consequence of morphological and cellular structure. However, the middle layer is the part with the most complex structure, containing a thick layer of VSMCs and a high elastine/collagen ratio [29]. Mature VSMCs are responsible for maintaining the circumference morphology of the aortic wall. Their elongated shape, which is characteristic for muscle tissue, allows the vessel to expand elastically, but it also can adapt to environmental stimuli and mechanical stress. Phenotype switch allows the VSMCs to go from contractile to synthetic, which is crucial in remodeling but contributes to vascular diseases when uncontrolled [30]. Contractile VSMCs support the contractile properties of the aortic wall and secrete specific muscle markers, as alpha smooth muscle actin (αSMA), smooth muscle 22α (SM22α) and calponin [31]. Synthetic VSMCs are more proliferative and migratory and synthetize a significant number of structural proteins. The elastine/collagen ratio increases in the proximal parts of the aorta to withstand high outlet blood pressure and decreases in the distal part, and the ratio diminishes, showing an increase in collagen [32]. Intima layer physiology is maintained by ECs. The cells prevent platelet activation, leukocyte adherence and blood clotting, though expressing factors like nitric oxide (NO) or prostacyclin (PGI2) [33]. Activated by inflammatory stimuli, ECs upregulate cellular adhesion molecules, including vascular cell adhesion molecule (VCAM) and intercellular adhesion molecule (ICAM), triggering leukocyte deposit on the endothelial surface [34]. The integrity and strength of the intima layer is crucial to maintaining the physiology of the circulatory system, which is also due to mechanical damages. Its interruption results in aortic dissection, during which blood flows between the layers of the aortic wall, as shown in Figure 2.

## 5. Abdominal Aorta Aneurysm—Cellular Physiology and Pathophysiology

The development of AAA and TAA differs slightly in pathophysiology. Factors predisposing to the development of AAA may include advanced age, family history and external factors, including lifestyle, that seem to have a major contribution to the occurrence of this disease. This contrasts with TAA, which is often attributed to single genetic defects, such as in Marfan syndrome or Loeys–Dietz syndrome. The above discrepancies may result from the various morphologies of TAA and AAA. The first difference may be the numerosity of elastin lamellae; along the aorta the number of elastic lamellae decreases. Subsequently, the elastin-to-collagen ratio in the abdominal segment of the aorta decreases [35]. Another aspect is the embryonic origin of SMCs. The thoracic aorta is surrounded by VSMCs of mesodermal origin, which are then replaced by a second wave of VSMCs that migrate from the neural crest. It is worth emphasizing that neural crest-derived VSMCs are presumably better suited to adaptively remodeling the thoracic aorta to withstand a higher pulse pressure and ejection volume by laying down more elastic lamellae during development. The abdominal aorta is settled only by VSMCs of mesodermal origin [36]. Additionally, the neural crest cell precursors of the thoracic aorta react to growth factors and cytokines in a different way from the mesodermal precursors of the abdominal aorta. An example may be homocysteine, which involves collagen deposition and stimulates the proliferation and synthetic activity of neural crest vascular smooth muscle cells, while cells of mesodermal origin remain unchanged [19]. Another example is transforming growth factor β (TGF-β), which is responsible for vascular development and regulates cell growth and differentiation. It has been shown that when treated with TGF-β, neural crest vascular SMCs demonstrated increased DNA synthesis and collagen production, whereas mesodermal vascular SMCs did not respond. This may explain why mutations in the TGF-β receptor may lead to TAA but have little effect on the abdominal aorta [37], although both diseases have similar symptoms and macroscopic mechanisms of action. AAA results from an accumulation of complex factors, shown in Figure 3. Overactive matrix metalloproteinases (MMPs) family proteins play a major role in AAA pathogenesis, especially MMP-1, MMP-2, MMP-9, MMP-12 and MMP-14 [38]. Furthermore, immunological response is participating, engaging signaling networks for macrophage migration inhibitory factor (MIF) and secreted phosphoprotein 1 (SPP1) [39]. Another hallmark of AAA is a strong inflammation, engaging immune cells as well as inflammatory pathway activation, like FOS and interleukine-17 (IL-17), interleukin-1β (IL-1β) and nuclear factor κB (NF-κB) [40,41]. Overgrown PVAT may fuel an inflammation in healthy VSMCs, increasing the level of proinflammatory factors with a decreased expression of anti-inflammatory molecules [42], inducing the infiltration of proinflammatory cells, as neutrophils, macrophages and lymphocytes [43]. The interplay of infiltrated inflammatory cells is complex in forming the inflammatory environment during AAA development. Lymphocyte T secretes cytokines that activate macrophages, which are polarized into two subgroups, contributing to the development of the disease in opposite ways [44]. M1 macrophages exacerbate local inflammatory processes and stimulate aortic wall degeneration. The stimulation of M2 macrophages occurs in the advanced stage of AAA and is involved in the processes of vascularization and counteracting ECM degradation and improves the repair processes of the aortic wall. Therefore, influencing the M1/M2 ratio of macrophages may have a therapeutic effect [45]. Antigen-presenting cells (APCs) such as dendritic cells, B lymphocytes and monocytes are also activated. Cellular hyperactivity results in the development of an immune response towards self-antigens, though AAA has hallmarks of an autoimmune disease [46]. Activating natural killer cells (NK cells) and mast cells during AAA pathogenesis suppress VSMCs’ viability, promoting the degradation of the aortic wall [47,48].

Tissue degradation is also preceded by necroptosis and apoptosis in both TAA and AAA [25]. The progression of AAA is strictly related to the up-regulation of apoptotic markers, i.e., Cas3 [49]. Deregulated expressions of miRNA level, as miR-520f-3p, miR-155-5p and miR-1-3p, were proposed as transcriptional biomarkers of AAA disease, among others, leading to apoptosis [50,51,52]. A specific type of cell death is ferroptosis. It is a type of programmed cell death dependent on excessive lipid peroxidation in high ironic conditions. It is characterized by mutual interference with NF-κB, inflammasome and MAPK signaling pathways [53]. Ferroptosis can be also induced by neutrophils extracellular traps (NETs), a system of self-defense mechanisms for immobilizing and neutralizing pathogens using a network loaded with bactericidal proteins, synthetized by neutrophils [54]. The decomposition of NETs or inhibiting ferroptosis significantly suppresses aneurysm development, indicating a high significance of these factors in inducing AAA pathogenesis [55,56]. A crucial apoptosis factor in AAA is oxidative stress, which is an excessive accumulation of reactive oxygen species (ROS). Cell death is the final stage of the cascade triggered by ROS, previously leading to an inhibition of cell metabolism and phenotype switching [57]. The role of VSMCs and aortic fibroblasts is, above all, synthetizing ECM components, though the loss of cells is strictly related to the loss of the ability to resynthesize degenerated collagen and elastin fibers.

ECM degradation is one of the main disease-causing agents of many aortic diseases [58,59]. Changes in collagen content in AAA are ambiguous; an increased collagen content improves aorta stiffness and susceptibility to rupture. However, a decreased collagen level can weaken the arterial wall structure, leading to AAA formation [60]. A decrease in elastin content, disrupting the collagen-to-elastin ratio balance, may influence the pathogenesis of various types of aneurysm and other aortic diseases, including atherosclerosis [61,62]. The degeneration of ECM component shows regional heterogeneity on the circumference of the aneurysm [63]. Genetic variants of synthesize collagen are also crucial for the mechanical properties of the aorta; heterozygous mutations in COL3A1 or COL5A1 are reported as factors determining weakened collagen matrix and a predisposition to AAA [64,65]. Mutations in the fibrillin-1 gene (FBN1) are the causative factor for Marfan syndrome (MFS), a disorder of connective tissue morphology of which one of the consequences is a fragmentation of elastin fibers, which may be a reason for MFS-related aortic aneurysm [66]. Mechanical stimulation, as wall stretching, is a physiological process of functioning vascular walls. Calcium deposits negatively correlate with wall elasticity, also up-regulating ECs’ expression of MMP-9, IL-6 and CD62e, a marker of macrophage adhesion [67].

The dysfunction of ECs, as cellular senescence, triggers an overproduction of cytokines/chemokines, reactive oxygen species (ROS) and cyclophilin A (CypA), inducing inflammatory cell infiltration, VSMCs apoptosis and ECM destruction, influencing the structure of the entire aortic wall [68]. Not only do the remodeling and degradation of ECM promote cell death and inflammation induced by MMP-12 [69] but inflammation and apoptosis also intensify ECM degradation [70]. Hallmarks of cellular senescence are markedly visible in the ECs and VSMCs of AAA patients [71]. The action of even one of the factors described above causes a cascade of degradation and a weakening of the aortic wall, weaving a complex network of AAA pathogenesis and development, influencing pathways of molecular factors, mentioned in Table 1.

## 6. Pathogenesis of Abdominal Aortic Aneurysm—Molecular Aspects

### 6.1. Inflammation—Markers

The predisposing factor to AAA occurs through an inflammatory process. Immune system cells, such as mast cells, macrophages, neutrophils and lymphocytes, penetrate through the layers of the aorta and produce inflammatory factors, such as interleukin-6 (IL-6), granulocyte macrophage colony-stimulating factor (GM-CSF), monocyte chemoattractant protein-1 (MCP-1), neutrophil gelatinase-associated lipocalin (NGAL), interferon-γ (IFN-γ) and homocysteine. The wide involvement of the immune system in the formation of AAA causes the recruitment and accumulation of immune cells to weaken the aortic wall and exterminate structurally important VSMCs/fibroblasts. This loss of wall content greatly reduces the mechanical resistance of the aorta. Immune cells also secrete proteases: chymase, tryptase and metalloproteinases (MMP), which induce ECM degradation and contribute to the widening and consequent rupture of the aorta [97].

#### 6.1.1. Chymase and Tryptase

Mast cells are cells of the connective tissue which participate in the inflammatory process, and they contribute to the formation of AAA by producing proteases and cytokines [98]. Chymase contributes to smooth muscle cell apoptosis as a result of SMC fibronectin degradation. The degradation of fibronectin enables the migration of inflammatory cells, leading to focal adhesion and consequently to the spread of inflammation in the aortic wall. Tryptase activates a protease-activated receptor 2 (PAR-2), affecting an increased permeability of the vessels, and PAR-2 activates phospholipase, which is membrane-bound and independent of A2 calcium (iPLA2). The phospholipase increases the adhesion of leukocytes to the endothelial cells and accelerates inflammation in the aorta, causing it to widen [75].

#### 6.1.2. Interleukin-6 (IL-6), Granulocyte Macrophage Colony-Stimulating Factor (GM-CSF) and Transforming Growth Factor Beta (TGF-β)

The most common proinflammatory cells are macrophages, which penetrate aortic layers and produce cytokines such as IL-6, TGF-β, GM-CSF [86]. IL-6 is a proinflammatory cytokine which contributes to the differentiation of monocytes to macrophages and stimulates the GM-CSF and signal transducer and activator of transcription 3 (STAT3) expression [40]. Moreover, its other function is initiating acute phase response, which is a non-specific response to the disorder in homeostasis resulting from infection, tissue damage and cancerous growth or immunological disorders. The signaling cascade starts when IL-6 becomes bound to the membrane receptor IL-6 (mIL-6R) and the subunit gp130. As a result, the balance between the synthesis and protein degradation of the substance in the intercellular aortic wall is disturbed [76]. TGF-β is involved in the regulation of growth and the differentiation of immune cells and is also important in CVDs. It is important to distinguish two signaling paths: the canonical TGF-β signaling pathway, which operates through Smad2 and/or Smad3 to transfer signals, and the non-canonical TGF-β signaling: ERK pathway [77], which activates extracellular signal-regulated kinase (ERK). ERK1/2 is an important upstream regulator of vascular smooth muscle cell (VSMC) phenotypic switch [99]. Each pathway accelerates the progression of AAA by widening the artery wall as a result of changing VSMCs’ phenotype, apoptosis and ECM weakening. TGF-α and TGF-β1 modulate both the PI3K/Akt and NFκB signaling pathways, which contribute to aneurysm development [100].

GM-CSF is a chemotactic cytokine encoded by Csf2, originally described to be involved in hematopoiesis, and it can stimulate stem cells to differentiate into immune cells [101]. GM-CSF is used as a drug to accelerate hematological recovery in anticancer responses [102]. Commonly used, and only approved by U.S. Food and Drug Administration, GM-CSF is named sargramostim (Leukine^®®^, Partner Therapeutics, Inc., Lexington, KY, USA) [103]. The effectiveness of GM-CSF can potentially be increased by use on a polymer scaffold. The slow release of the drug from the implant does not cause cytotoxicity and ensures long-lasting drug action [104]. Additionally, several findings showed defects in the upregulation of GM-CSF contributing to diseases of the circulatory system such as AAA. GM-CSF induces macrophage infiltration into the aortic wall during inflammation, as well as the activation of M1 macrophages. The M1 take part in the production of proteolytic enzymes and proinflammatory cytokines, such as IL-6 and TNF, that contribute to the formation of AAA.

#### 6.1.3. Monocyte Chemoattractant Protein-1 (MCP-1)

MCP-1 is a chemotactic cytokine involved in the pathogenesis of CVDs including AAA. Cytokine is encoded by the CCL2 gene, and its high expression is commonly related to the fibrosis of various organs, as liver or lungs [105,106]. MCP-1 participates in the progression of vasculitis through the activation of the FAK-MCP-1 axis. Focal adhesion kinase (FAK or PTK2) is a cytoplasmic tyrosine kinase involved in the transmission of a signal, which modulates cellular mobility contributing to the recruitment and influx of macrophages into the wall of the aorta [79]. An overexpression of MCP-1 is incorporated into the expression of C-C chemokine receptor type 2 (CCR2), which is an MCP-1 receptor, intensifying the influence of its effect. Other upregulated genes are lysozyme 1 (Lyz1), which is antibacterial factor, and immune cell surface markers CD52 and CD180 [78].

#### 6.1.4. Reactive Oxygen Species (ROS)

The main side effect of cellular metabolism is ROS production, molecular oxygen derivatives whose crucial sources are redox processes in mitochondria [107]. A physiological ROS level is necessary for cell signaling and regulating cellular metabolism [108]. The management of ROS is tightly controlled by the antioxidant system, containing proteins such as superoxide dismutase (SOD) and glutathione peroxidase (GPx). A disturbed antioxidant system and an increased rate of uncontrolled ROS release result in its accumulation, called oxidative stress (OS). The synthesis of collagens, including collagen I, which is most abundant in the ECM, is probably regulated by ROS, but research is still ongoing. Liu et al. used human uterosacral ligament-derived fibroblasts to form a cell-coated scaffold, as a model of collagen degradation [109]. The structure was treated with a low concentration of H_2_O_2_ to induce OS, which led to a decrease in COL1A1 (gene codifying for collagen type I α1 chain) expression. However, increasing the concentration of H_2_O_2_ results in an increase in COL1A1 expression, suggesting that mild OS occurs with normal metabolism or that minor injuries may contribute to promoting collagen. In the case of wound healing, collagen deposition is required; however, it may also be responsible for tissue damage, e.g., fibrosis. Various studies have shown that increased levels of ROS cause acceleration in levels of collagen I and III, contributing to heart muscle fibrosis [110].

Fibronectin (FN) also influences oxidative stress. FN expression has been shown to be positively regulated in the presence of different sources of oxidative stress [111]. The source of ROS in the vessels are inflammatory cells, which penetrate through the vessel wall. In addition, ROS contribute to the reduction in NO availability which leads to endothelium dysfunction. At the same time, there is an acceleration in adhesion molecules such as ICAM-1 (intercellular adhesion molecule 1) and interlukin-1α, which enables a progressive migration of inflammatory cells, e.g., lymphocytes and macrophages [80]. The reduced availability of NO, together with ROS’ influence on proinflammatory cells, increases the level of cytokine expression, which sustains the inflammation process in the aortic wall [112]. ROS play a significant role in regulating the phenotypic switch and proliferation of VSMCs. In vascular tissue, this process involves the signaling pathway of Megakaryocytic leukemia 1 (MKL1), E2 promoter binding factor 1 (E2F1) and forkhead box protein M1 (FOXM1) [82].

#### 6.1.5. Neutrophil Gelatinase-Associated Lipocalin (NGAL)

NGAL is an acute phase protein which is found in granularities of neutrophilic granulocytes and is a potential marker of AAA development and progression. NGAL expression in the aorta, in its middle layer, is correlated with caspase-3 expression (enzymes of the cysteine proteases group) being an apoptosis marker. As a result of that process, VSMCs of the aorta are characterized by increased apoptosis [83].

Other evidence for the promotion of apoptosis by NGAL is the activation of caspase 3, 8 and 9 in nasopharyngeal carcinoma (NPC) [113]. Nevertheless, regarding AAA, NGAL expression also predisposes to the formation of NGAL/MMP-9 complexes and consequently protects MMP-9 from proteolytic degradation causing the degradation of the ECM components of the aorta [84].

#### 6.1.6. Interferon-γ (IFN-γ)

IFN-γ is a type of cytokine which has a proinflammatory effect. IFN-γ affects the course of inflammatory reactions, secreting, among others, an IFN-γ-inducible protein 10 (CXCL10). This protein promotes the adhesion of T cells to the endothelium and also takes part in the activation and chemotaxis of macrophages, thus contributing to the onset of an inflammatory process which affects the formation of AAA [76]. IFN-γ secreted by activated T cells promotes local ECM degradation through the recruitment of macrophages that are engaged in MMP production, especially MMP-2 and MMP-9 [85]. The activity of IFN-γ also impairs endothelial cell metabolism. Vascular ECs produce much of their energy depending on glycolysis. That process is disturbed by IFN-γ-dependent tryptophan catabolism destabilization and depleting nicotinamide adenine dinucleotide [114]. Mitochondria injury induced by IFN-γ leads to apoptosis due to the decline in mitochondrial membrane potential and the promoting of endoplasmic reticulum (ER) stress [115]. This contributes to accelerated cell senescence and ECs apoptosis [116], accelerating the development of vascular diseases and injuries.

#### 6.1.7. Homocysteine

Homocysteine is an amino acid produced in the hepatic metabolism of methionine and is involved in the pathogenesis of AAA. The excessive accumulation of this compound leads to an imbalance of lymphocytes T CD4^+^ (glycoprotein found on Th/helper T lymphocyte). As a consequence, endothelial damage and elastin degradation occur in the aortic wall. Homocysteine exhibits cytotoxic properties which disrupt the functioning of the vascular endothelium. Therefore, in conditions of elevated homocysteine, endothelial cells stop producing NO, and thus dysfunction in its production also leads to the increased activity of the inflammatory process and smooth muscle cell proliferation [117]. Homocysteine also induces the expression of a wide range of metabolic factors, such as autotaxin (ATX). Upregulated endothelial ATX expression recruits T-cells into the aortic walls, inducing vascular inflammation and consequently accelerating the development of AAA [118].

#### 6.1.8. Metalloproteinases (MMP)

The most common cause of AAA is the degradation of ECM by MMP, which enables the movement of cells during inflammation as well as in normal physiological processes [119]. ECM fills the space between cells, and its main components are collagen and elastin. These ECM proteins present in the aortic wall are responsible for the mechanical properties of the aorta. Elastic fibers provide elasticity, whereas collagen has a stretching effect on the aortic wall. ECM is also involved in signal transmission between the outer environment as well as the growth and migration of immune system cells. Thus, it plays a significant role in the non-physiological reconstruction of the aortic wall [119]. The AAA-weakened wall is characterized by the reduced production of elastin and collagen, as well as the reduced expression of metalloproteinase inhibitors. Elastin is a structural protein, a component of elastic fibers that belongs to the main components of ECM, which results in it being a key building protein in the arteries. Elastin accounts for about 30% of the weight of the artery. This protein provides flexibility and elasticity and allows for the deformation of blood vessels, and thus, together with collagen, it determines the passive mechanics of the abdominal aorta. Its other functions include the regulation of the proliferation, adhesion, migration and organization of the smooth muscle cell cytoskeleton. It also has chemotactic effects on macrophages [60]. One of the causes of AAA formation may be the degradation of elastin fibers by increasing the amount and activity of enzymes belonging to different classes, including cysteine proteases (cathepsin), serine proteases (e.g., neutrophil and pancreatic elastase) and proteins from the family of metalloproteinases of the extracellular matrix, in particular MMP-1, MMP-2 and MMP-9. The metalloproteinases have a significant effect on the condition of elastic fibers in the arteries. In addition, an increase in the expression of these proteins in degraded elastic plaques has been demonstrated. The next factor in the reduction in the expression of MMP inhibitors is TIMPs (Tissue Inhibitors of MMPs) [120].

Another component of ECM that plays an important role in creating AAA is collagen. This protein is stabilized by transverse bonds, and as a result, collagen fibers are characterized by significant mechanical strength and are not susceptible to stretching. Unlike elastin, collagen fibers are less stretchable. In addition, stretching collagen too strongly causes irreversible structural changes and a loss of strength. Changes in collagen metabolism disturb its fibrinogenesis and cause the remodeling of ECM in the aortic wall. In addition, increasing mechanical load and deformation ultimately leads to exceeding AAA wall stability and leads to rupture, resulting in high mortality [121]. Proteolytic enzymes play an important role in ECM homeostasis in healthy tissues as well as in the formation of aneurysms. MMPs lead to the splitting of collagen fibers and thus weaken the aortic wall [87]. MMPs are proteolytic enzymes that belong to zinc- and calcium-dependent endopeptidases that can degrade ECM (collagen, elastin, etc.) and also participate in processes such as inflammatory response CVDs and cancer infiltrates [122]. These enzymes, under physiological conditions, are involved in tissue remodeling, tissue differentiation and cell migration. MMPs are produced by macrophages, fibroblasts, vascular smooth muscle cells and endothelial cells. Homeostatic disorder in the matrix structure leads to an imbalance between MMPs and their tissue inhibitors. These inhibitors are tissue-specific inhibitors of matrix metalloproteinase whose function is to inhibit MMP proteolytic activity [123]. ECM components such as collagen and elastin play a significant role in the integrity of the aortic wall and serve as important substrates for MMP. These enzymes may degrade type I-X and type XIV collagen and other ECM components, including elastin. The structural loss of ECM occurs by preventing the adhesion of ECM components. This process contributes to the degradation of ECM, and thus enables the formation of AAA [124]. Metalloproteinase 1 (MMP-1) is an enzyme involved in the proteolysis of collagen fibers. An increased expression of the MMP-1 gene is a marker of the occurrence of abdominal aortic aneurysm. Macrophages produce MMPs such as MMP-2, MMP-3 and MMP-12. MMP-2 is synthesized by fibroblasts and smooth muscle cells, and changes in the MMP-2 structure lead to pathological changes in the vessels and the formation of small aneurysms. MMP-3 (stromelizine-1) is produced by epithelial tissue cells and fibroblasts. Increased expression is a risk factor for AAA by an initiating and promoting the infiltration and release of inflammatory cell processes. It is also believed that MMP-3 may activate other MMPs. MMP-9 (type IV) collagenase is a multifunctional proteolytic protease affecting collagen and elastin. As a result of this process, type IV collagenase contributes to the degradation of ECM proteins in the aortic wall [88]. Another factor in the formation of AAA may be COL4A1/A2, found in the artery wall. COL4A1 and COL4A2 form the basal membranes and are present in every tissue of the body. In addition, they provide tissue strength. The expression of COL4A1/A2 in the artery, showing strong anti-permeable activity, may play a protective role in preventing the migration of inflammatory cells to the aortic wall. Thus, reduced levels of COL4A1/A2 in the aortic artery increase the likelihood of AAA formation [125]. Under physiological conditions, VSMCs will not produce MMP-9, but inflammation in the aortic wall will cause excessive an expression of MMP-9. As a result, the elastin is damaged, and AAA is formed. Elastin degradation products contribute to increased leukocyte chemotaxis by implying the inflammation of the aortic wall. In addition, MMP-2 and MMP-9 can activate TNF and TGF-β, stimulating the growth of cancer cells [126]. MMP-12 can degrade elastin and plays a key role in macrophage migration. MMP-12 expression is increased in the case of AAA [127]

#### 6.1.9. Lysine Oxidase (LOX)

Another marker contributing to the formation of AAA is LOX. It is an enzyme responsible for the cross-linking of collagen and elastin in the extracellular space, and it is produced by VSM cells. LOX inhibitors prevent collagen and elastin cross-linking. The destabilization of the aortic wall causes it to expand and eventually leads to the formation of an aneurysm [89]. LOX expression increases significantly in ANG-II-induced AAA, suggesting mechanisms preventing disease formation. The effect of ANG-II is enhanced when β-aminopropionitrile (β-APN), a LOX inhibitor, is present, resulting in undisturbed AAA development [90].

#### 6.1.10. Osteoprotegerin (OPG)

OPG is another marker of AAA. This compound is a cytokine from the TNF family. It is also responsible for bone homeostasis. The severity of AAA may be reduced by OPG via inhibition tumor necrosis factor-related apoptosis-inducing ligand (Trail). Trail induces MMPs secretion and also its own expression in VSMCs, triggering c-Jun-NH(2)-terminal kinase (JNK) signaling pathway [91]. OPG deficiency results in the hyperactivation of the Trail-induced Jnk-MMP9 pathway and AAA progression [90]. Its other function is the secretion of MMP-2 and MMP-9, which via endothelial cells, monocytes and VSMC degrade ECM [128]. OPG also contributes to the interaction of the control receptor activator of nuclear factor-κB (RANK) and its ligand (RANKL). Though this process, OPG triggers the NF-κB activation pathway [75].

#### 6.1.11. Osteopontin (OPN)

OPN is a phosphorylated glycoprotein secreted by VSMC. This glycoprotein is involved in bone growth and calcification [94]. In addition, it is involved in inflammation by the degranulation of mastocytes and is also a chemoattractant for neutrophils, participates in proteolysis and has the ability to inhibit the expression of TR1 lymphocytes (regulatory lymphocytes), which is a strong suppressor of inflammation that intensifies degeneration in the aorta wall. It also interacts with proinflammatory cytokines such as interleukine 1-β and TNF, which increase OPN expression. One of the main AAA pathophysiological factors, Ang II, is also the major factor increasing the expression of OPN [86]. In addition, plasma OPN concentrations are correlated with the development of CVDs, including AAA [129]. The decreased expression of OPN is associated with the non-contractile osteochondrogenic VSMCs phenotype, which is one of the AAA hallmarks, supporting calcification, hyperchondroplasia and aneurysm development [93].

#### 6.1.12. Cathepsins

Cathepsin belongs to lysosomal proteins that are activated in an acidic environment. AAA risk factors such as smoking contribute to the degradation of endothelial cells, leading to lysosomal membrane permeability, which leads to the secretion of cathepsins [11], whose production is regulated by changes in pH. However, cathepsin secretion is also regulated by cystatin C. The main cytokines responsible for the production of cathepsins are TNFα and INF-γ in vascular ECs, VSMCs and macrophages [96]. Cathepsin participates in the reconstruction of ECM by the degradation of their structural components (including collagen, elastin) and also regulates chemotaxis and angiogenesis via ERL chemokines (glutamate-leucine-arginine). Chemokines can influence macrophage migration to the aortic wall by inducing inflammation progression and as a result contribute to the formation of AAA [95].

### 6.2. VSMCs

AAA pathophysiology is characterized by VSMC apoptosis. The components of the mid-aortic layer are mainly VSMCs, which are involved in maintaining the AAA structure by controlling ECM secretion and proliferation, and they ensure the proper regulation of blood pressure. Stressors within the aorta wall contribute to the infiltration of inflammatory cells and change the effect of VSMCs. This process causes VSMC apoptosis as a result of the activity of cytokines such as MCP-1 protein. Another function of these cells is to ensure balance in switching the phenotype of the aortic wall from contractile to synthetic. The inflammatory process induces the change in the VSMC phenotype to synthetic by the expression of CD137, which is a transmembrane protein belonging to the superfamily of TNF receptors. Phenotypic switching occurs through direct binding to NFATc1 (nuclear factor of activated T cells, cytoplasmic 1), which inhibits the expression of VSMC contractile protein [130]. As a result, there is less VSMC contractility in the synthetic phenotype, and the marker of this change is a reduced expression of 22-alpha smooth muscle protein (SM22alpha) [131]. SM22alpha is a protein specific for smooth muscles, and combined with actin, it is involved in maintaining the structure of the aortic walls [132]. In addition, there is an increased expression of MMPs which degrade the ECM of the aorta and thus enable VSMCs’ proliferation and aortic expansion [133].

### 6.3. Hypertension and Abdominal Aortic Aneurysm

Arterial hypertension damages the aortic wall, making it less elastic and therefore more susceptible to AAA formation [134]. Kobeissi et al. include a meta-analysis of 21 cohort studies (20 publications) covering 28,162 AAA patients among 5,440,588 participants. The meta-analysis showed a 66% higher risk of AAA in patients with hypertension compared to patients without hypertension [135]. Hypertension is one of the inductive factors of vascular remodeling. Vessels remodeling under the conditions occur with abnormal VSMC proliferation and distorted collagen synthesis [136]. Long-lasting hypertension also upregulates the expression of factors contributing to AAA, like MAPK, PI3K/AKT pathways and FOXM1 [137,138]. Spontaneous hypertension influences the TGF-β1/Smad signaling pathway, taking part in forming fibrotic changes, contributing also to AAA pathophysiology [139].

### 6.4. Genetic Factors

There is a genetic aspect to the etiology of an aneurysm. The first studies in which the genetic basis of aneurysms was observed appeared in 1977: specifically, a case report of three brothers operated on due to ruptured aneurysms was published [140]. However, this disease may affect second-degree relatives [141].

The prevalence of TAA inheritance is approximately 20% [142]. Hannuksela et al. investigated seven first families referred to the Center for Cardiovascular Genetics at Umeå University Hospital with TAA. They showed that about 20% of patients with TAA have a first-degree relative with a similar disease [143]. In the genetic context, it should also be emphasized that TAA has two subtypes: syndromic and non-syndromic. Syndromic TAA involves other organ systems in addition to the aorta, whereas non-syndromic TAA is confined to the aorta. The syndromic subtype is linked to the dysfunction of the ECM, medial smooth muscle cells (SMC) or TGF-β signaling, namely, there is a genetic syndrome that affects the aorta and causes other changes in the body. Genetic syndromes that can lead to aortic disease include MFS, Loeys–Dietz syndrome, Turner syndrome and Vascular Ehlers–Danlos syndrome [144]. MFS is an autosomal dominant disease affecting connective tissue. Organ changes in MFS affect the entire body, including blood vessels. The syndrome is caused by a mutation in the fibrillin-1 (FBN1) gene, with a mutation frequency of 12.7% [145]; furthermore, a mutation in this gene may occur in AAA with a frequency of 5% [146]. FBN1 encodes the protein fibrillin-1 [147]. Fibrillin-1 is an ECM protein that forms polymers called microfibrils that tightly bind to elastic fibers. As a result, this disease reduces the strength and integrity of, among others, blood vessels such as the aorta with a high content of elastin [148]. Moreover, fibrillin-1 affects the proper binding of TGF-β protein in connective tissue. In the physiological state, TGF-β is captured by fibrillin into connective tissue. In this disease, the TGF-β protein is not bound by fibrillin and remains as a free particle in the blood, which leads to the abnormal behavior of connective tissue cells. As a result, the aortic wall is weakened and may be predisposed to the formation of TAA [149]. Loeys–Dietz syndrome (LDS) is a congenital multisystem connective tissue disease, inherited in an autosomal dominant manner, characterized by, among others, the occurrence of TAA. The above disease is caused by a mutation in the transforming growth factor type 1 or 2 gene (TGFBR1, TGFBR2), with mutation observed in appropriately 7%, 1% [150] and in AAA frequency of mutation TGFBR1—3% TGFBR2—2% [146]. TGF-β signaling plays an important role in blood vessel development and influences vascular homeostasis and maintenance [151]. Vascular Ehlers–Danlos syndrome (VEDS) is an autosomal dominant genetic disease affecting connective tissue. The cause of the disease is pathogenic variants in COL3A1, leading to defective or reduced type III collagen production [152], with a mutation frequency in AAA of 2% [146].

Non-syndromic TAA represents a heterogeneous group wherein there is a familial clustering of aortic disease, typically presenting at a young age in the absence of syndromic features. At least genes have been identified, generally encoding proteins involved in the extracellular matrix, vascular smooth muscle cell function or TGF-β signaling. Genes associated with TAA include smooth muscle α2 actin (ACTA2), forkhead box protein E3 (FOXE3), myosin-11 (MYH11) and myosin light chain kinase (MYLK) [153]. ACTA2 encodes the SMC isoform of α-actin. ACTA2 missense mutations were observed in appropriately 16% AAA patients [154]. ACTA2 disrupts α-actin polymerization and leads to the decreased contractility of aortic SMC; as a result, it causes predisposition to the development of TAA [155]. FOXE3 encodes a forkhead transcription factor (FOXO), which controls cell cycle and apoptosis. However, as a result of FOXE3 mutations, the apoptosis of aortic SMC occurs, and ultimately with increased pressure rupture may occur [131]. The next gene may be MYH11 with prevalence of 1% in TAA cases [156] and 7% in AAA [146]. This gene encodes smooth muscle myosin heavy chain (SM-MHC), a contractile component produced by SMC. The MYH11 gene mutation is involved in the dysregulation of SMC contraction, which is critical for maintaining aortic wall stability, leading to a susceptibility to TAA [157]. Myosin light chain kinase (MLCK), encoded by MYLK, is another gene that has been linked to non-syndromic TAA, but it also appears in AAA. The mutation in the MYLK gene occurs with a frequency of 12% in the case of AAA [146]. MYLK encodes myosin light chain kinase which is a calcium/calmodulin dependent enzyme. This kinase phosphorylates myosin regulatory light chains to facilitate myosin interaction with actin filaments to produce contractile activity. A mutation in the MYLK gene reduces MLCK activity, resulting in impaired contraction SMC in the aorta. A mutation in the MYLK gene reduces MLCK activity, resulting in impaired SMC contraction in the aorta [158]. The prevalence of inherited AAA was examined by Luijtgaarden et al., who analyzed patients presented with AAA at the vascular surgical outpatient clinic of the Erasmus University Medical Centre for AAA between 2004 and 2012. They indicated that a familial occurrence of AAA was reported by 128 of the 568 index AAA patients, pointing to a prevalence of 22.5% [15]. Another example of research could be the study covering the screening of siblings with AAA in Sweden by Linné et al. They examined 338 index AAA patients and indicated a 9.8% prevalence of Familial AAA [159]. Sakalihasan et al. investigated patients with AAA diagnosed at the Cardiovascular Surgery Department, University Hospital of Liege, Belgium, between 1999 and 2012. The number of AAA index cases was 144, indicating a 12.9% prevalence of familial AAA [160]. Linné et al. screened siblings in Stockholm, Sweden and showed that the prevalence of familial AAA was 10.6% [159]. SMAD4 (mothers against decapentaplegic homolog 4), also known as DPC4 (deleted in pancreatic cancer-4), is member of the SMAD protein family [161]. SMAD4 is required in VSMCs to ensure a normal aortic structure. Da Ros et al. showed that the deletion of SMAD4 in VSMCs increased the level of IL-1b, which is a key mediator of the inflammatory response, causing predisposition to the development of AAA [162]. A reduced level of GM3, a type of glycosphingolipid, created in ferroptosis, is noticed in AAA development. Additionally, the silencing GM3 synthase gene (ST3GAL5) disturbs iron accumulation in VSMCs, promoting AAA development [163]. A high expression of angiopoietin-like protein 8 (ANGPTL8) is observed in progressive AAA, whereas the knockout of that gene efficiently reduces AAA formation via reducing cell apoptosis, inflammatory cytokines and MMPs expression [164]. The current state of knowledge does not allow for effective drugs for the prevention of AAA development to be obtained; however, the newest literature involves attempts to define drug-target relationship therapy. Single-cell RNA sequencing and Network Medical Framework were implemented by Niu & Wang to discover mentioned relationships. Screening the differences between AAA and nonaneurysmal cells has identified two proteins, Solute Carrier Family 2 Member 3 (SLC2A3) and Immediate Early Response 3 (IER3), as a potential key targets strictly related to inflammation and immune response [165]. A potentially pointed therapeutic compound is DB08213, factor binding to SLC2A3 and inhibiting its activity [165].

## 7. Conclusions

AAA has a complex pathogenesis, but inflammation is the most common pathological factor. The development of the inflammatory process is enabled by proteases secreted by mastocytes, such as chymase and tryptase, which contribute to the migration of inflammatory cells through the degradation of fibronectin and increase the adhesion of leukocytes to endothelial cells.

The inflammatory process is also enabled by cytokines, such as IL-6, TGF-β, GM-CSF, MCP-1 and IFN-γ, as a result of the infiltration of the aortic wall. Another marker of aortic inflammation may be homocysteine, which contributes to the progression of aortic inflammation, leading to T cell imbalance T CD4^+^. OPN, a phosphorylated glycoprotein secreted by VSMCs, reduces the expression of TR1 lymphocytes (regulatory lymphocytes), also increasing inflammation. Therefore, changes in the structure of the aortic wall are characterized by degradations in ECM structural elements such as collagen and elastin and VSMCs apoptosis. ECM degradation occurs through the activity of cathepsins and metalloproteinases affecting collagen and elastin. An increased expression of LOX inhibitors is also characterized, which is responsible for the cross-linking of collagen and elastin in ECM. Aortic VSMC apoptosis also occurs through the action of metalloproteinases by NGAL found in neutrophils and ROS. In addition, homeostasis in VSMCs is disturbed, which is manifested by switching from the shrink phenotype to synthetic VSMCs, increasing VSMC proliferation and thus widening the aortic wall. In addition, type IV collagen (COL4A1/A2) found in the artery wall may be a marker with protective effects against the abdominal aortic artery. The expression of this protein has an anti-permeable effect and thus prevents the migration of inflammatory cells to the layers of the aorta.

The rupture of an abdominal aortic aneurysm has a high mortality rate, but its detection is a clinical problem. As a result, it is necessary to conduct significant research on the causes of AAA, which may bring clinicians closer to the early detection of this disease.

Understanding the mechanisms leading to the development of AAA not only allows for a more effective therapy but, more importantly, offers the patient a more detailed diagnosis, enabling earlier, less costly and less damaging treatments.

The development of a holistic understanding of the pathogenesis of aneurysms, which remains a clinical problem, is the subject of numerous studies focusing on identifying symptoms as early as possible, aiming to know the possible risks before the development of clinical symptoms.

## Figures and Tables

**Figure 1 cells-13-00274-f001:**
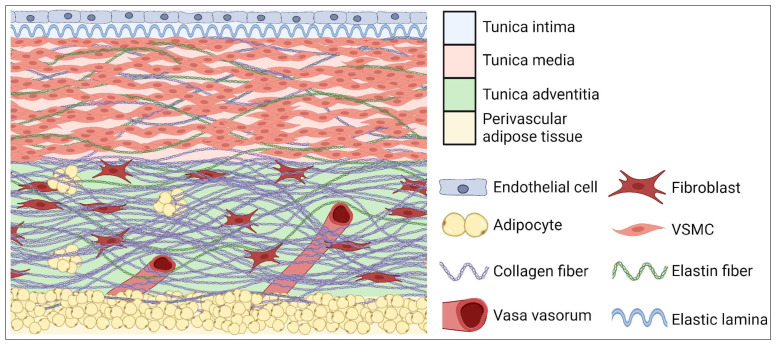
Layers of aortic wall.

**Figure 2 cells-13-00274-f002:**
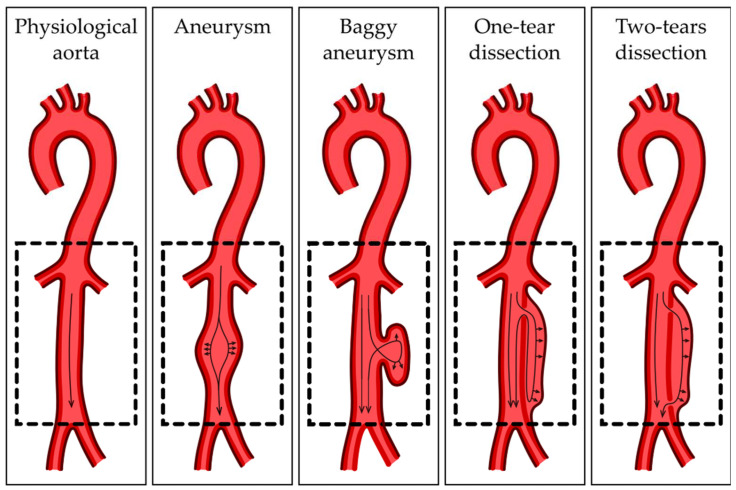
Types of abdominal aorta aneurysms. Arrows show direction of blood pressure.

**Figure 3 cells-13-00274-f003:**
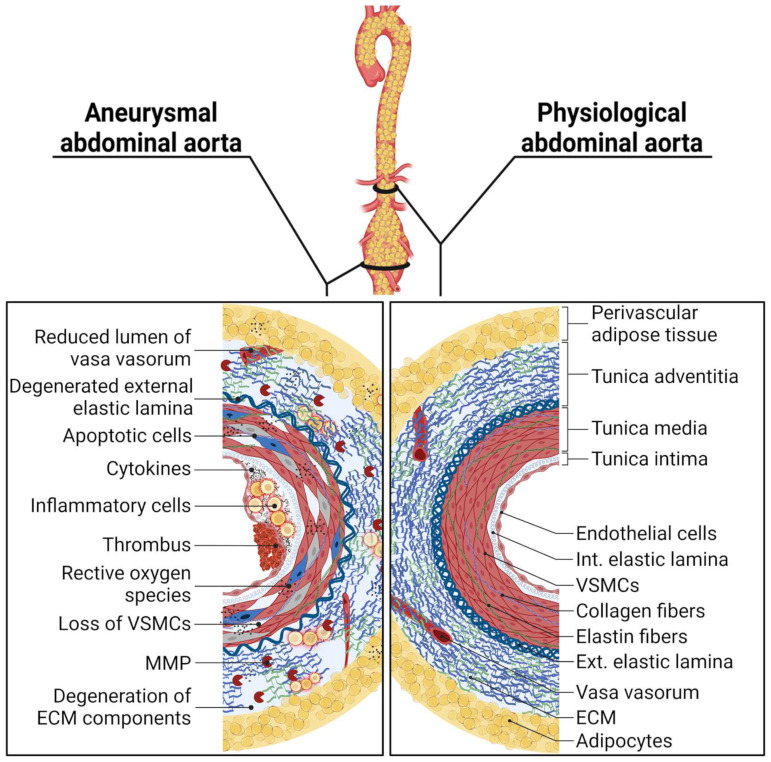
Molecular comparison of physiological and aneurysmal abdominal aorta (Abbreviations: VSMCs, vascular smooth muscle cells; MMP, matrix metalloproteinase; ECM, extracellular matrix; Int., internal; Ext., external).

**Table 1 cells-13-00274-t001:** Molecular factors involved in pathogenesis of abdominal aortic aneurysm.

Factor	Molecular Interactions	Pathophysiological Significance	References
Chymase	Transform pro-MMP-9 to active form; induce TNF and ANG-II expression	Chymase inhibitors suppress the accumulation of vascular tumor cells, reduce mean blood pressure and total renal blood flow on murine model	[72,73]
Tryptase	Activate PAR-2 and iPLA2; induce LTC4, PGD2 expression	Tryptase is upregulated in several CVDs, inducing the fibrosis, inflammation, calcification and permeability of the vessels	[74,75]
IL-6	Stimulate GM-CSF and STAT3 expression	IL-6 signaling accumulates proteolytically-active macrophages, inducing vascular inflammation	[40,76]
GM-CSF	Upregulate MCP-1 expression	A deficiency of GM-CSF inhibits AAA and leads to a decrease in inflammatory monocytes, as well as the activation of M1 macrophage	[22]
TGF-β	Upregulate ERK expression	Accelerates AAA progression by widening the artery wall, though VSMC proliferation	[77]
MCP-1	Upregulate CCR2, Lyz1, Cd52, Cd180 and other inflammatory/immune cell markers	MCP-1 pathways contribute to T cell receptor signaling, chemokine signaling, phagocytosis and inflammatory-related pathways	[78,79]
ROS	Upregulate MKL1, E2F1, FOXM1, ICAM-1 and plenty of cardiovascular pathophysiological factors	ROS neutralization reduces inflammatory cell infiltration, VSMC migration, phenotype switching and apoptosis	[80,81,82]
NGAL	Upregulate Cas-3 and protect MMP-9 against degradation	NGAL blood concentration indicates vessel wall deterioration. Its level is even higher in patients with ruptured AAA	[83,84]
IFN-γ	Upregulate CXCL10, MMP-2 and MMP-9	Plays as an inflammatory factor, enhances endothelium adhesion and macrophage recruitment	[76,85]
HCY	Activate Ang-II receptor, upregulate ATX (or ENPP2) and IL-αL, -β1, -β2 and -β7	Induces blood clotting, the degradation of elastin in the elastic membrane and the processes of fibrosis and calcification	[48,86]
MMPs	Can affect the expression of other types of MMPs. Some of them are engaged in increasing TNF and TGF-β levels.	MMPs are directly responsible for the degradation of ECM components and thus the pathophysiology of AAA	[87,88]
LOX	Expression of LOX is related with ANG-II induced AAA; its activity is inhibited by β-APN.	The role of LOX is to enhance the cross-linking of collagen and elastin, contributing to the integrity and stabilization of a healthy vascular wall.	[89,90]
OPG	OPG deficiency enhances the trail-induced Jnk-MMP9 pathway, controls RANK/RANKL interactions	A decreased level of OPG expression constitutes an AAA marker.	[75,91,92]
OPN	Inhibit the expression of TR1 lymphocytes, chemoattract the inflammatory cells	It is strictly related to VSMC phenotype switch, supporting calcification, hyperchondroplasia and aneurysm development	[86,93,94]
Cathepsins	Its expression is regulated by TNFα and INF-γ; regulates the distribution of ERL chemokines	Regulates the chemotaxis of macrophages, influencing the development of AAA and promoting angiogenesis	[95,96]

Abbreviations: MMP, matrix metalloproteinase; TNF, Tumor necrosis factor; ANG-II, Angiotensin Type II; PAR-2, Protease-activated receptor 2; iPLA2, calcium-independent phospholipase A2; LTC4, Leukotriene C4; PGD2, Prostaglandin D2; IL, Interleukine; GM-CSF, Granulocyte macrophage colony stimulating factor; STAT3, Signal transducer and activator of transcription 3; TGF-β, Tumor growth factor β; ERK, Extracellular signal-regulated kinase; MCP-1, Monocyte chemoattractant protein 1; CCR2, C-C chemokine receptor type 2; Lyz1, Lysozyme 1; ROS, Reactive oxygen species; MKL1, Megakaryoblastic leukemia 1; E2F1, E2 promoter binding factor 1; FOXM1, Forkhead Box M1; ICAM-1, Intercellular adhesion molecule 1; NGAL, Neutrophil gelatinase-associated lipocalin; Cas-3, Caspase 3; IFN-γ, Interferon gamma; CXCL10, C-X-C motif chemokine ligand 10; HCY, Homocysteine; ATX (ENPP3), Autotaxin (Ectonucleotide Pyrophosphatase/Phosphodiesterase 2); LOX, Lysyl oxidase; β-APN, beta-aminopropionitrile fumarate; OPG, Osteoprotegerin; Trail, Tumor necrosis factor–related apoptosis-inducing ligand; Jnk, c-Jun NH2-terminal kinase; RANK, receptor activator of nuclear factor-κB; RANKL, Receptor activator for nuclear factor κB Ligand; OPN, Osteopontin; TR1, T regulatory type 1.

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
