# Peer review of "Cellular, Molecular and Clinical Aspects of Aortic Aneurysm—Vascular Physiology and Pathophysiology"

_cells, 2024, doi:10.3390/cells13030274_

Round 1
Reviewer 1 Report
Comments and Suggestions for Authors
This paper reviewed the current study status of AAA, and be helpful for understanding the pathogenesis of this lethal aortic disease.
Minor comment.
1. Line 36. “Abdominal aortic aneurysm is a” should be “Abdominal aortic aneurysm (AAA) is a”. An aberration should be given, and check all similar mistakes, such as line 48. MMPs.
2. Line 80-81 “middle” before media and “outer” before adventitia are not necessary.
3. Fig.1 and 3, much elastin in adventitia. Please revise this mistake.
4. Whether hypertension also contribute to AAA pathogenesis, you may need to discuss it in “conclusion” or “perspective”.
Author Response
13.12.2023
MSc. Dominika Domagała
Division of Anatomy
Department of Human Morphology and Embryology
Wroclaw Medical University, Wroclaw, Poland
Dear Editor
revised version of the manuscript entitled: Cellular, molecular and clinical aspects of aortic aneurysm – vascular physiology and pathophysiology has been uploaded for your further consideration.
Thank you for the opportunity to revise and resubmit this manuscript. I am grateful for the valuable time and meticulous feedback provided by the reviewers. I have integrated the reviewer’s suggested changes into the manuscript. In the revised manuscript, you will find the incorporated changes clearly coloured in red, marked with comments. Below, you will find a point-by-point response addressing each of your comments and concerns.
We hope that these revisions brings the manuscript into a form appropriate for publication in your journal.
Sincerely,
Dominika Domagała
Reviewer comment:
This paper reviewed the current study status of AAA, and be helpful for understanding the pathogenesis of this lethal aortic disease.
Minor comment.
- Line 36. “Abdominal aortic aneurysm is a” should be “Abdominal aortic aneurysm (AAA) is a”. An aberration should be given, and check all similar mistakes, such as line 48. MMPs.
Response 1: This correction will clarify terms of issues contained in the article. We added the abbreviations in main text of article.
“Abdominal aortic aneurysm (AAA) is a clinically challenging and potentially lethal disease that often escapes early detection, and…”
“AAA has a multifaceted etiology, including genetic factors, as high expression of matrix metalloproteinases (MMPs) [6], cytokines…”
- Line 80-81 “middle” before media and “outer” before adventitia are not necessary.
Response 2: We have corrected these statements to simplify the manuscript text.’
“The media is made up of smooth muscle, elastin and collagen fibers which make it possible for the vessel to deform. The adventitia constitutes mainly of…”
- 1 and 3, much elastin in adventitia. Please revise this mistake.
Response 3: We have incorporated your suggestion throughout the manuscript. Collagen fibers constitutes the vast majority of the fiber content in the aortic adventitia, as considered in both figures.
- Whether hypertension also contribute to AAA pathogenesis, you may need to discuss it in “conclusion” or “perspective”.
Response 4: This aspect of AAA pathophysiology contributes significantly to the development of the disease. Placing it in the body of the manuscript will significantly improve its quality and expand the context of the content. We have included information about hypertension as a separate paragraph.
“6.3. Hypertension and abdominal aortic aneurysm
Arterial hypertension damages the aortic wall, making it less elastic and therefore more susceptible to AAA formation [131]. Kobeissi et al. include a meta-analysis 21 cohort studies (20 publications) covering 28,162 AAA patients among 5,440,588 participants. A meta-analysis showed a 66% higher risk of AAA in patients with hypertension compared to patients without hypertension [132]. Hypertension is one of the inductive factor of vascular remodeling. Vessels remodeling under the conditions occur with abnormal VSMCs proliferation and distorted collagen synthesis [133]. Long-lasting hypertension also upregulate expression of factors contributing AAA, like MAPK, PI3K/AKT pathways and FOXM1 [134,135]. Spontaneous hypertension influe on TGF-β1/Smad signaling pathway, taking a part in forming fibrotic changes, contributing also on AAA pathophysiology [136].”
Reviewer 2 Report
Comments and Suggestions for Authors
The authors have collected a very large number of research studies of diverse mechanism proposed in the etiology of AAA. With a emphasis on the immunologic involvement.
Genetic factors are a major cause of AAA and TAA. At least 20% of the AAA and TAA patient report familial aneurysms, when the families are screened for aneurysms, risk for relatives is sharply increased compared to the population, and 5% of cases have a pathogenic variant. Indicating that yet unknown genetic susceptibilities play an important role in the etiology.
The review lacks information on the so far identified aneurysm genes, their (inter) action and role in the initiation of AAA. t a review would nowadays be expected to present this information and -in addition- show discuss how the effects of genetic defects includes interaction with the mechanism presented in the paper. For instance like the effect of SMAD4 gene defects on interleukin expression ( PMID: 29150241)
Overall: more emphasis is needed that none of the presented etiologies is sufficient, that AAA is a complex disorder involving interactions between different mechanisms and genetic susceptibilities.
In addition:
line 68, ref 17 seems inappropriate
line 83: the media does contains collagen and elastin
line 98: what is the disproportion between thoracic and abdominal aorta?
line 104 and 114; not correct!: in the proximal thoracic aorta there are a large number - up to 30-40 - of elastine layers in the media, to withstand high pressure. more distally in the aorta the number of elastine layers decreases, in favor of the collagen/ elastine ratio.
line 127: unclear : no explanation is given why TAA and AAA should have different pathophysiology . Genetic aneurysm show otherwise: both AAA and TAA can be caused by same genetic defect. in heritable forms of aneurysm: co occurrence of TAA and AAA in one patient occurs and we see also that within a family some relatives have AAA while others have TAA. These observations do not support this section of the manuscript.
In addition authors should include the studies that refute the hypothesis that AAA is driven by atherosclerosis since many patients do not have atherosclerosis.
Comments on the Quality of English Language
good and clearly written
Author Response
13.12.2023
MSc. Dominika Domagała
Division of Anatomy
Department of Human Morphology and Embryology
Wroclaw Medical University, Wroclaw, Poland
Dear Editor
I have uploaded a revised version of the manuscript entitled: Cellular, molecular and clinical aspects of aortic aneurysm – vascular physiology and pathophysiology. Thank you for giving the opportunity to submit a revised draft of manuscript. Your commitment to reviewing and offering valuable feedback on my manuscript is greatly appreciated. I have incorporated changes in response to the queires clearly coloured in red. Below, you will find a point-by-point response addressing each of your comments and concerns.
Yours sincerely,
Dominika Domagała
Reviewer comment:
The authors have collected a very large number of research studies of diverse mechanism proposed in the etiology of AAA. With a emphasis on the immunologic involvement.
Major comments
- Genetic factors are a major cause of AAA and TAA. At least 20% of the AAA and TAA patient report familial aneurysms, when the families are screened for aneurysms, risk for relatives is sharply increased compared to the population, and 5% of cases have a pathogenic variant. Indicating that yet unknown genetic susceptibilities play an important role in the etiology.
The review lacks information on the so far identified aneurysm genes, their (inter) action and role in the initiation of AAA. t a review would nowadays be expected to present this information and -in addition- show discuss how the effects of genetic defects includes interaction with the mechanism presented in the paper. For instance like the effect of SMAD4 gene defects on interleukin expression ( PMID: 29150241)
Overall: more emphasis is needed that none of the presented etiologies is sufficient, that AAA is a complex disorder involving interactions between different mechanisms and genetic susceptibilities.
Response 1: The genetic aspect of AAA has been added in the following passage as directed by the reviwer.
“An important aspect of the CVDs diagnosis is the history of diseases in the family, this also applies to AAA. Genetic factors contribute to the development of AAA, especially in first-degree relatives [139]. As states by statistical screening, male descendants of parents suffering CVDs have higher risk of inheriting AAA predispositions [140]. The two strongest genes supporting evidence for a genetic risk contribution to AAA, namely the CDKN2BAS gene known as ANRIL, encodes an antisense RNA that regulates the expression of cyclin-dependent kinase inhibitors CDKN2A and CDKN2B and DAB2IP, which encodes an inhibitor of cell growth and survival [141]. Another example is SMAD4 (mothers against decapentaplegic homolog 4) also known as DPC4 (deleted in pancreatic cancer-4) is member of the SMAD protein family [142]. SMAD4 is required in VSMCs to ensure normal aortic structure. Da Ros et al. showed that deletion of SMAD4 in VSMCs increased the level of IL-1b, which is a key mediator of the inflammatory response, pre-disposing to the development of AAA [143]. In AAA development, is noticed reduced level of GM3, a type of glycosphingolipid, contributed in ferroptosis. Also, silencing GM3 synthase gene (ST3GAL5) disturb an iron accumulation in VSMCs, promoting AAA development [144]. High expression of angiopoietin-like protein 8 (ANGPTL8) is observed in progressive AAA, whereas knockout of that gene efficiently reduce AAA forming, via reducing cell apoptosis, inflammatory cytokines and MMPs expression [145].The current state of knowledge does not allow obtaining effective drugs to prevent the development of AAA, however, in the newest literature are attempts to define drug-target relationship therapy. Single-cell RNA sequencing and Network Medical Framework were implement-ed by Niu & Wang to discover mentioned relationships. Screening the differences between AAA and nonaneurysmal cells identify two proteins, Solute Carrier Family 2 Member 3 (SLC2A3) and Immediate Early Response 3 (IER3), as a potential key targets, strictly related with inflammation and immune response [146]. Potentially pointed therapeutic compound is DB08213, factor binding to SLC2A3 and inhibiting its activity [146]”
- Line 68, ref 17 seems inappropriate.
Response 2: It has been corrected in manuscript.
“Yuan, Z.; Lu, Y.; Wei, J.; Wu, J.; Yang, J.; Cai, Z. Abdominal Aortic Aneurysm: Roles of Inflammatory Cells. Front. Immunol. 2021, 11, 609161, doi:10.3389/fimmu.2020.609161.”
- Line 83: the media does contains collagen and elastin.
Response 3: We have added the following statements in response to the reviewers concern.
- Line 98: what is the disproportion between thoracic and abdominal aorta?
Response 4: The disproportion refers to incidence. We have changed the term from disproportion to incidence in the text of the manuscript.
- Line 104 and 114; not correct!: in the proximal thoracic aorta there are a large number - up to 30-40 - of elastine layers in the media, to withstand high pressure. more distally in the aorta the number of elastine layers decreases, in favor of the collagen/ elastine ratio.
Response 5: The sentence has been corrected. “Elastin content in whole human aortic ECM is less than collagen, especially in distal parts of aorta [33]. Elastin-to-collagen content increases in proximal parts of aorta to withstand high outlet blood pressure”.
- Line 127: unclear : no explanation is given why TAA and AAA should have different pathophysiology . Genetic aneurysm show otherwise: both AAA and TAA can be caused by same genetic defect. in heritable forms of aneurysm: co occurrence of TAA and AAA in one patient occurs and we see also that within a family some relatives have AAA while others have TAA. These observations do not support this section of the manuscript.
Response 6: Thank you for drawing attention to this issue. Of course TAA and AAA have slightly different reports of pathophysiology. Nevertheless, both pathophysiologies are very complex and complicated, depending on many factors, so it is difficult to determine exactly what is going on. TAA is described as a disease with a greater genetic basis and inheritance, but the current literature still requires development and comprehensive research.
“The development of AAA and TAA differ slightly in pathophysiology, although both diseases have similar symptoms and macroscopic mechanism of action”.
- In addition authors should include the studies that refute the hypothesis that AAA is driven by atherosclerosis since many patients do not have atherosclerosis.
Response 7: The manuscript has undergone a comprehensive rebalancing. We emphasized the fact that that further research on AAA and atherosclerosis is necessary.
“Disturbances in the communication of aortic layers, like depositions of atherosclerotic plaque between layers may be one of the factors contributing to the development of AAA. Atherosclerotic plaques consequently reduces the lumen of the vessels and significantly impedes blood flow through it [28]. As a result, AAA may be formed as a reaction to histopathological changes. The narrowing of the aortic lumen causes compensatory changes in the aortic wall and, consequently, ECM remodeling occurs to normalize the vessel lumen diameter. Rapidly occurring remodeling significantly reduce the organized structure of individual layers, weakening the strength and elasticity of the vessels [29]. However, further research is needed regarding the above relationship”.
Round 2
Reviewer 2 Report
Comments and Suggestions for Authors
Unfortunately the authors have not been able to address the comments accurately. The additions regarding the genetics of aortic aneurysm are still insufficient and also incorect.
Genetic factors contribute to the development of AAA, especially in first-degree relatives [139]. genetic causes usually affect close relatives most, because these share a larger number of genes, with the index case, compared to more distant relatives. This does not preclude that second degree relatives can be affected.
This statement does not provide any insight into the genetic susceptibilities for aneurysm, that these present with syndromic and nonsyndromic genetic forms of aneurysm in about 1:3 patients.
As states by statistical screening, male descendants of parents suffering CVDs have higher risk of inheriting AAA predispositions [140].
The term statistical screening does not apply in this context: there have been different kind of studies to show familial aggregation; family screening studies and family history studies. For example https://doi.org/10.1177/1358863X16686409 see suppl table S 2

The two strongest genes supporting evidence for a genetic risk contribution to AAA, namely the CDKN2BAS gene known as ANRIL, encodes an antisense RNA that regulates the expression of cyclin-dependent kinase inhibitors CDKN2A and CDKN2B and DAB2IP, which encodes an inhibitor of cell growth and survival [141].
These two examples are presented,incorrectly as strong aneurysm genes. These are- on the contrary: loci identified by genetic association studies, and therefore there is no functional evidence that defects in these genes have any effect on the aortic wall and can cause an aneurysm.
On the contrary there are now some 50 or more genes that have been identified as aneurysm genes, and are tested in diagnostic setting. The FBN1 gene, is the best know, for it is associated with Marfan syndrome. Online many overveiuw of aneurys gene can be found, for instance https://doi.org/10.1016/j.jacc.2018.04.089
The discovery of these genes have laid the way to identify that the TGFB signaling pathway has a major and important role in AAA pathogenesis.
Response 5: The sentence has been corrected. “Elastin content in whole human aortic ECM is less than collagen, especially in distal parts of aorta [33]. Elastin-to-collagen content increases in proximal parts of aorta to withstand high outlet blood pressure”.
It would be more accurate to focus on the difference in the elastine/collagen ratio which is is high in the proximal aorta and lower in the distal part: where this ratio diminishes showing an increase in collagen.
Author Response
Thank you for your review, the answer is attached.
Sincerely,
Dominika Domagała

Round 3
Reviewer 2 Report
Comments and Suggestions for Authors
48-50
AAA has a multifaceted etiology, including genetic factors, as high expression of MMPs [6], cytokines [7], microRNAs 49 (miRNA) [8] or single nucleotide variants (SNV) of individual genes, as MMP-2, MMP-3, 50 MMP-13 [9], IL-6 and TNF-α [10].
Complex multifactorial etiology is the correct term instead of multifaceted.
The rest of the sentence is also confusing :
- since high expression of MMP or cytokines cannot be the etiology, the etiology is de reason why there is a high expression of MMp or cytokine etc.
- is unclear what is the difference between genetic factors and SNV in individual genes?
68: what is involution change
81 the importance is that the aorta retains its elasticity and contractility. The statement that is needs to be able to deform is incorrect
136 =138 I the statement that AAA and TAA are different requires references that indicate what are the precise differences and what is the origin of these differences
The new paragraph on genetics contains was a necessary improvement but has some major errors:
Timp has been associated with cerebral aneurysm and not with aortic aneurysm, as far as we know .
Turner syndrome should be presented as a chromosomal defect not as a mendelian inherited disorder
The whole first paragraph of the section on genetic factors should be rewritten to focus on
1. Prevalence of inherited cq familial disease in AAA and in TAA. Use the tables that I have send previously to find the correct references.
2. The prevalence of mutations in aneurysm genes in AAA and in TAA. Look for references
3. Gender differences in outcome or clinical presentation should not be presented as a genetic trait.
528 provide references of specific studies that show that this statement holds
Comments on the Quality of English Language
not applicable
Author Response
|
27.01.2024 |
MSc. Dominika Domagała
Division of Anatomy
Department of Human Morphology and Embryology
Wroclaw Medical University, Wroclaw, Poland
Dear Editor
I have uploaded a revised version of the manuscript entitled: Cellular, molecular and clinical aspects of aortic aneurysm – vascular physiology and pathophysiology. Thank you for giving me the opportunity to submit a revised draft of the manuscript. Your thoughtful suggestions have helped us improve the quality of my manuscript. Below, you will find a point-by-point response addressing each of the comments and concerns.
Yours sincerely,
Dominika Domagała
Reviewer comment:
- 48-50
AAA has a multifaceted etiology, including genetic factors, as high expression of MMPs [6], cytokines [7], microRNAs 49 (miRNA) [8] or single nucleotide variants (SNV) of individual genes, as MMP-2, MMP-3, 50 MMP-13 [9], IL-6 and TNF-α [10].
Complex multifactorial etiology is the correct term instead of multifaceted
Response 1: Revision made as suggested by the Reviewer/
“AAA has a complex multifactorial etiology….”
The rest of the sentence is also confusing:
- since high expression of MMP or cytokines cannot be the etiology, the etiology is de reason why there is a high expression of MMP or cytokine etc.
Response 2: We removed that statement.
- is unclear what is the difference between genetic factors and SNV in individual genes?
Response 3: Our intention was to list the factors that are taking a part in a etiology because there are no pointed differences.
- 68: what is involution change
Response 4: Advanced age.
“Involutional changes (advanced age)”
- 81 the importance is that the aorta retains its elasticity and contractility. The statement that is needs to be able to deform is incorrect
Response 5: We deleted “deform” and we added importance is that the aorta retains its elasticity and contractility
“The media is composed of smooth muscle, elastin, and collagen fibers which provides aorta elasticity and contractility”
- 136 =138 I the statement that AAA and TAA are different requires references that indicate what are the precise differences and what is the origin of these differences
Response 6: We added the following passage in respons to the reviewer’s concerns.
The development of AAA and TAA differ slightly in pathophysiology. The factors predisposing to the development of AAA may include: advanced age, family history and external factors, including lifestyle, that seems to have a major contribution to occurrence of this disease. It is contrasting to TAA which often is attributed to single genetic defects, such as in Marfan syndrome or Loeys-Dietz syndrome. The discrepancies may result from various morphology of TAA and AAA. The first difference may be numerosity of elastin lamellae, along the aorta the number of elastic lamellae decreases. Subsequently the elastin-to-collagen ratio in abdominal segment of aortic in which it is decreased (He et al., 2022).Another aspect is embryonic origin of SMCs. The thoracic aorta is surrounded by SMCs of mesodermal origin, then are replaced by a second wave of SMCs that migrate from the neural crest. It is worth emphasizing that neural crest–derived SMCs presumably more well suited to adaptively remodel the thoracic aorta to withstand the higher pulse pressure and ejection volume by laying down more elastic lamellae during development. In the abdominal aorta is settled only by SMCs of mesodermal origin. (Allaire et al., 2009). Additionally, the neural crest cell precursors of the thoracic aorta react in a different way to growth factors and cytokines than the mesodermal precursors of the abdominal aorta. The example may be homocysteine, which involved collagen deposition, stimulates the proliferation and synthetic activity of neural crest vascular smooth muscle cells, while cells of mesodermal origin remain unchanged (Sun et al., 2018). Another example is TGF-β, which is responsible for vascular development and regulator of cell growth and differentiation. It has been shown that treated with TGF-β, neural crest vascular SMCs demonstrated increased DNA synthesis and collagen production, whereas mesodermal vascular SMCs did not respond. Whereby, this may explain that why mutations in the TGF-β receptor may lead to TAA, but have little effect on the abdominal aorta (Kuivaniemi et al., 2015).
Allaire, E., Schneider, F., Saucy, F., Dai, J., Cochennec, F., Michineau, S., Zidi, M., Becquemin, J. P., Kirsch, M., & Gervais, M. (2009). New Insight in Aetiopathogenesis of Aortic Diseases. In European Journal of Vascular and Endovascular Surgery (Vol. 37, Issue 5, pp. 531–537). https://doi.org/10.1016/j.ejvs.2009.02.002
He, B., Zhan, Y., Cai, C., Yu, D., Wei, Q., Quan, L., Huang, D., Liu, Y., Li, Z., Liu, L., & Pan, X. (2022). Common molecular mechanism and immune infiltration patterns of thoracic and abdominal aortic aneurysms. Frontiers in Immunology, 13. https://doi.org/10.3389/fimmu.2022.1030976
Hiratzka, L. F., Bakris, G. L., Beckman, J. A., Bersin, R. M., Carr, V. F., Casey, D. E., Eagle, K. A., Hermann, L. K., Isselbacher, E. M., Kazerooni, E. A., Kouchoukos, N. T., Lytle, B. W., Milewicz, D. M., Reich, D. L., Sen, S., Shinn, J. A., Svensson, L. G., Williams, D. M., Jacobs, A. K., … Yancy, C. W. (2010). 2010 ACCF/AHA/AATS/ACR/ASA/SCA/SCAI/SIR/STS/SVM guidelines for the diagnosis and management of patients with thoracic aortic disease: Executive summary: A report of the american college of cardiology foundation/american heart association task force on practice guidelines, american association for thoracic surgery, american college of radiology, american stroke association. In Circulation (Vol. 121, Issue 13). Lippincott Williams and Wilkins. https://doi.org/10.1161/CIR.0b013e3181d4739e
Kuivaniemi, H., Ryer, E. J., Elmore, J. R., & Tromp, G. (2015). Understanding the pathogenesis of abdominal aortic aneurysms. In Expert Review of Cardiovascular Therapy (Vol. 13, Issue 9, pp. 975–987). Taylor and Francis Ltd. https://doi.org/10.1586/14779072.2015.1074861
Proost, D., Vandeweyer, G., Meester, J. A. N., Salemink, S., Kempers, M., Ingram, C., Peeters, N., Saenen, J., Vrints, C., Lacro, R. V., Roden, D., Wuyts, W., Dietz, H. C., Mortier, G., Loeys, B. L., & Van Laer, L. (2015). Performant Mutation Identification Using Targeted Next-Generation Sequencing of 14 Thoracic Aortic Aneurysm Genes. Human Mutation, 36(8), 808–814. https://doi.org/10.1002/humu.22802
Sun, J., Deng, H., Zhou, Z., Xiong, X., & Gao, L. (2018). Endothelium as a potential target for treatment of abdominal aortic aneurysm. In Oxidative Medicine and Cellular Longevity (Vol. 2018). Hindawi Limited. https://doi.org/10.1155/2018/6306542
van de Luijtgaarden, K. M., Heijsman, D., Maugeri, A., Weiss, M. M., Verhagen, H. J. M., IJpma, A., Brüggenwirth, H. T., & Majoor-Krakauer, D. (2015). First genetic analysis of aneurysm genes in familial and sporadic abdominal aortic aneurysm. Human Genetics, 134(8), 881–893. https://doi.org/10.1007/s00439-015-1567-0
- The new paragraph on genetics contains was a necessary improvement but has some major errors:
Timp has been associated with cerebral aneurysm and not with aortic aneurysm, as far as we know .
Turner syndrome should be presented as a chromosomal defect not as a mendelian inherited disorder
Response 7: We have removed the statements that the reviewr
The whole first paragraph of the section on genetic factors should be rewritten to focus on
- Prevalence of inherited cq familial disease in AAA and in TAA. Use the tables that I have send previously to find the correct references.
- The prevalence of mutations in aneurysm genes in AAA and in TAA. Look for references
- Gender differences in outcome or clinical presentation should not be presented as a genetic trait.
Response 8: We have moved gender differences into epidemiology as suggested by the reviewer .
Response 9: We added to our section on genetic factor: Prevalence of inherited cq familial disease and we used tables as suggested by the reviewerTAA
„…….Prevalence of TAA inheritance is approximately 20% (Rodrigues Bento et al., 2022). Hannuksela et al. (2015) investigated seven first families referred to the Center for Cardiovascular Genetics at Umeå University Hospital with TAA. Showed that about 20% of patients with TAA have a first-degree relative with a similar disease (Hannuksela et al., 2015)”
AAA
„…Prevalence of inherited AAA were examined by Luijtgaarden et al., and analyzed patients presented with AAA at the vascular surgical outpatient clinic of the Erasmus University Medical Centre for AAA between 2004 and 2012. They indicated that familial occurrence of AAA was reported by 128 of the 568 index AAA patients, pointing a prevalence of 22.5% (Van De Luijtgaarden et al., 2017). Another example of research could be the study covering screening of siblings with AAA in Sweden by Linné et al.. They detected of the 338 index AAA patients and indicated 9.8% of Familial AAA (Linné et al., 2016). Sakalihasan et al. have investigated patients with AAA diagnosed at the Cardiovascular Surgery Department, University Hospital of Liege, Belgium, between 1999 and 2012. The AAA index cases was 144 and indicated 12.9% familial AAA (Sakalihasan et al., 2014). Linné et al. screened siblings in Stockholm, Sweden and showed that prevalence of familial AAA was 10.6% (Linné et al., 2012)”
Response 10: We added to our section on genetic factor: The prevalence of mutations in aneurysm genes in AAA and in TAA.
„In the genetic context it should also be emphasized that TAA, has two subtypes: syndromic and non-syndromic. Syndromic TAA involves other organ systems in addition to the aorta, whereas non-syndromic TAA is confined to the aorta. In case of syndromic subtype is linked to dysfunction of the ECM, SMCs, or TGF-β signaling, namely there is a genetic syndrome that affects the aorta and causes other changes in the body. Genetic syndromes that can lead to aortic disease include: MFS, Loeys-Dietz syndrome, Turner syndrome, Vascular Ehlers-Danlos syndrome [143]. MFS is an autosomal dominant disease affecting connective tissue. Organ changes in MFS affect the entire body, including: blood vessels. The syndrome is caused by a mutation in the fibrillin-1 (FBN1) gene, with a mutation frequency of 12.7% (Proost et al., 2015),also a mutation in this gene may occur in AAA with a frequency of 5% (van de Luijtgaarden et al., 2015)……”
„Loeys-Dietz syndrome (LDS) is a congenital multisystem connective tissue disease, inherited in an autosomal dominant manner, characterized by among others, the occurrence of TAA. The above disease is caused by a mutation in the transforming growth factor type 1 or 2 gene (TGFBR1, TGFBR2) with mutation observed in appropriately 7%, 1% (De Cario et al., 2018) and in AAA frequency of mutation TGFBR1 – 3% TGFBR2 – 2% (van de Luijtgaarden et al., 2015)……”
“Vascular Ehlers-Danlos syndrome (VEDS) is an autosomal dominant genetic disease affecting connective tissue. The cause of the disease is pathogenic variants in COL3A1, leading to defective or reduced type III collagen production [149], with a mutation frequency in AAA of 2% (van de Luijtgaarden et al., 2015)”
“ACTA2 encodes of the SMCs isoform of α-actin. ACTA2 missense mutations with observed in appropriately 16% AAA patients (Renard et al., 2013)”
“The next gene may be MYH11with prevalence of 1% in TAA cases (Hiratzka et al., 2010)and 7% in AAA(van de Luijtgaarden et al., 2015) “
“A mutation in the MYLK gene with a frequency of 12% in the case of AAA (van de Luijtgaarden et al., 2015)”
- 528 provide references of specific studies that show that this statement holds
Response 11: We removed this statement as suggested by the reviewer:
We have removed this statement, because after analyzing the publications: inheritance appears more often in the case of AAA:
Klarin, D., Damrauer, S. M., Tsao, P. S., Verma, S. S., Judy, R., Dikilitas, O., Wolford, B. N., Paranjpe, I., Levin, M. G., Pan, C., Tcheandjieu, C., Spin, J. M., Lynch, J., Assimes, T. L., Åldstedt Nyrønning, L., Mattsson, E., Edwards, T. L., Denny, J., Larson, E., … Ritchie, M. (2020). Genetic Architecture of Abdominal Aortic Aneurysm in the Million Veteran Program. Circulation, 142(17), 1633–1646. https://doi.org/10.1161/CIRCULATIONAHA.120.047544
„Abdominal aortic aneurysm (AAA) is a complex disease affected by both environmental1
And genetic factors,2 and heritability has been estimated to be as high as 70% (Klarin et al., 2020)”.
Singh, T. P., Field, M. A., Bown, M. J., Jones, G. T., & Golledge, J. (2021). Systematic review of genome-wide association studies of abdominal aortic aneurysm. In Atherosclerosis (Vol. 327, pp. 39–48). Elsevier Ireland Ltd. https://doi.org/10.1016/j.atherosclerosis.2021.05.001
„Abdominal aortic aneurysm (AAA) is an important cause of death worldwide and has an
estimated heritability between 70 and 77%”
Allaire, E., Schneider, F., Saucy, F., Dai, J., Cochennec, F., Michineau, S., Zidi, M., Becquemin, J. P., Kirsch, M., & Gervais, M. (2009). New Insight in Aetiopathogenesis of Aortic Diseases. In European Journal of Vascular and Endovascular Surgery (Vol. 37, Issue 5, pp. 531–537). https://doi.org/10.1016/j.ejvs.2009.02.002
He, B., Zhan, Y., Cai, C., Yu, D., Wei, Q., Quan, L., Huang, D., Liu, Y., Li, Z., Liu, L., & Pan, X. (2022). Common molecular mechanism and immune infiltration patterns of thoracic and abdominal aortic aneurysms. Frontiers in Immunology, 13. https://doi.org/10.3389/fimmu.2022.1030976
Hiratzka, L. F., Bakris, G. L., Beckman, J. A., Bersin, R. M., Carr, V. F., Casey, D. E., Eagle, K. A., Hermann, L. K., Isselbacher, E. M., Kazerooni, E. A., Kouchoukos, N. T., Lytle, B. W., Milewicz, D. M., Reich, D. L., Sen, S., Shinn, J. A., Svensson, L. G., Williams, D. M., Jacobs, A. K., … Yancy, C. W. (2010). 2010 ACCF/AHA/AATS/ACR/ASA/SCA/SCAI/SIR/STS/SVM guidelines for the diagnosis and management of patients with thoracic aortic disease: Executive summary: A report of the american college of cardiology foundation/american heart association task force on practice guidelines, american association for thoracic surgery, american college of radiology, american stroke association. In Circulation (Vol. 121, Issue 13). Lippincott Williams and Wilkins. https://doi.org/10.1161/CIR.0b013e3181d4739e
Kuivaniemi, H., Ryer, E. J., Elmore, J. R., & Tromp, G. (2015). Understanding the pathogenesis of abdominal aortic aneurysms. In Expert Review of Cardiovascular Therapy (Vol. 13, Issue 9, pp. 975–987). Taylor and Francis Ltd. https://doi.org/10.1586/14779072.2015.1074861
Proost, D., Vandeweyer, G., Meester, J. A. N., Salemink, S., Kempers, M., Ingram, C., Peeters, N., Saenen, J., Vrints, C., Lacro, R. V., Roden, D., Wuyts, W., Dietz, H. C., Mortier, G., Loeys, B. L., & Van Laer, L. (2015). Performant Mutation Identification Using Targeted Next-Generation Sequencing of 14 Thoracic Aortic Aneurysm Genes. Human Mutation, 36(8), 808–814. https://doi.org/10.1002/humu.22802
Sun, J., Deng, H., Zhou, Z., Xiong, X., & Gao, L. (2018). Endothelium as a potential target for treatment of abdominal aortic aneurysm. In Oxidative Medicine and Cellular Longevity (Vol. 2018). Hindawi Limited. https://doi.org/10.1155/2018/6306542
van de Luijtgaarden, K. M., Heijsman, D., Maugeri, A., Weiss, M. M., Verhagen, H. J. M., IJpma, A., Brüggenwirth, H. T., & Majoor-Krakauer, D. (2015). First genetic analysis of aneurysm genes in familial and sporadic abdominal aortic aneurysm. Human Genetics, 134(8), 881–893. https://doi.org/10.1007/s00439-015-1567-0
Thank you for considering our request to publish the article, and we hope that the extensive changes we have made in the text of the manuscript have brough it into a form that is acceptable for publication.